# An AC-Rich Bean Element Serves as an Ethylene-Responsive Element in *Arabidopsis*

**DOI:** 10.3390/plants9081033

**Published:** 2020-08-14

**Authors:** Chunying Wang, Tingting Lin, Mengqi Wang, Xiaoting Qi

**Affiliations:** College of Life Science, Capital Normal University, Beijing 100048, China; 2190801018@cnu.edu.cn (C.W.); zoetingy@gmail.com (T.L.); mitchkeyblues@gmail.com (M.W.)

**Keywords:** AC-rich element, ethylene-responsive element, ethylene response factor, *Arabidopsis*

## Abstract

Ethylene-responsive elements (EREs), such as the GCC box, are critical for ethylene-regulated transcription in plants. Our previous work identified a 19-bp AC-rich element (ACE) in the promoter of bean (*Phaseolus vulgaris*) *metal response element-binding transcription factor 1* (*PvMTF-1*). Ethylene response factor 15 (PvERF15) directly binds ACE to enhance *PvMTF-1* expression. As a novel ERF-binding element, ACE exhibits a significant difference from the GCC box. Here, we demonstrated that ACE serves as an ERE in *Arabidopsis*. It conferred the minimal promoter to respond to the ethylene stress and inhibition of ethylene. Moreover, the *cis*-acting element ACE could specifically bind the nuclear proteins in vitro. We further revealed that the first 9-bp sequence of ACE (ACE_core_) is importantly required by the binding of nuclear proteins. In addition, *PvERF15* and *PvMTF-1* were strongly induced by ethylene in bean seedlings. Since PvERF15 activates *PvMTF-1* via ACE, ACE is involved in ethylene-induced *PvMTF-1* expression. Taken together, our findings provide genetic and biochemical evidence for a new ERE.

## 1. Introduction

Gas hormone ethylene regulates numerous development processes and stress responses in plants [1]. The control of these processes by ethylene involves the complex regulation of ethylene biosynthesis, ethylene perception, and signal transduction. A conserved ethylene signaling pathway has been established in model plants such as *Arabidopsis* (*Arabidopsis thaliana*) and rice (*Oryza sativa*) [2]. In the ethylene-induced transcription cascade, the ethylene response factor (ERF) is a crucial component that binds to the GCC box in the promoters of target genes to activate the ethylene response [3]. The GCC box (5′-AGCCGCC-3′) was originally identified in the tobacco (*Nicotiana* tabacum) β-1,3-glucanase *Gln2* gene and is thought to be a common ethylene-responsive element (ERE) [4]. The GCC box is widely contained in the promoter regions of ethylene-inducible defense genes, but not in the regulatory regions of fruit-ripening genes and flower petal senescence genes [5]. Ethylene-responsive promoter regions without the GCC box have been identified in the senescence-related carnation glutathione-S-transferase gene [6] and fruit-ripening tomato *E4* gene [7], indicating that the GCC box may not be the only ERF-binding element in the process of ethylene-regulated transcription.

Our previous work described a 19-bp AC-rich element (ACE, 5′-CCTAAACCCCAAAACAATC-3′) in the promoter of bean (*Phaseolus vulgaris*) *metal response element-binding transcription factor 1* (*PvMTF-1*) [8]. The bean ethylene response factor 15 (PvERF15) directly binds ACE to activate *PvMTF-1* expression [8]. Interestingly, this newly identified ERF-binding element, ACE, exhibits different properties from the GCC box. Thus, it is of great interest to further investigate whether ACE acts as an ERE. In this study, we characterized ACE as a new functional ERE in *Arabidopsis*. Our findings extend our knowledge of ethylene-regulated transcription.

## 2. Results

### 2.1. ACE Causes the Minimal Promoter to Respond to Ethylene in Arabidopsis Seedlings

To check whether ACE acts as an ERE in plants, we performed the transient β-glucuronidase (GUS) expression assay in the etiolated *Arabidopsis* seedlings. Ethylene stress was achieved by applying the ethylene precursor 1-aminocyclopropane-1-carboxylic acid (ACC) [9], whereas endogenous ethylene perception was blocked with Ag^+^ (an inhibitor of the ethylene receptors) [10]. Results showed that 10 μM ACC treatment produced a typical ethylene-responsive phenotype. It significantly reduced the hypocotyl or root length of the etiolated *Arabidopsis* seedlings (Figure 1A). However, the Ag^+^ (from 50 μM AgNO_3_)-treated etiolated *Arabidopsis* seedlings significantly increased in hypocotyl length due to the inhibition of endogenous ethylene action (Figure 1A). These observations are consistent with previous studies [11].

Next, these seedlings were transfected with the GUS reporter construct (Figure 1B), in which *GUS* was driven by two copies of ACE or mutated ACE (5′-CCTAAACCCCAAAACAATC-3′ mutated to 5′-ttTAAAttttAAAAtAATt-3′, named mACE [8]) fused at the upstream of the minimal cauliflower mosaic virus 35S promoter (*35S_(−46/+8)_*). A *35S:GFP* construct was used as an internal control by co-transformation to normalize the transformation efficiency in each experiment. We found that for *Arabidopsis* seedlings transfected with *ACE-35S_(−46/+8)_:GUS,* the GUS activity was induced by approximately 2-fold in the presence of ACC, while AgNO_3_ treatment decreased the GUS activity by about 40% (Figure 1C). A significant change in GUS activity was not observed in the plants transfected with *mACE-35S_(−46/+8)_:GUS* (Figure 1C), suggesting that ACE lost its ethylene responsiveness due to the mutation. These results showed that ACE allows *35S_(−46/+8)_* to respond to both the ethylene stress and inhibition of ethylene in the etiolated *Arabidopsis* seedlings. We also noted that plants expressing *ACE-35S_(−46/+8)_:GUS* exhibited higher GUS accumulation than plants expressing *mACE-35S_(−46/+8)_:GUS* (Figure 1C, control lanes). This observation is consistent with previous studies that demonstrate that ACE acts as a positive element in transient GUS assays [8].

### 2.2. ACE Specifically Binds to the Nuclear Proteins of Arabidopsis

Next, we performed electrophoretic mobility shift assay (EMSA) to determine whether ACE interacts specifically with the etiolated *Arabidopsis* nuclear proteins. The nuclear proteins were immunologically verified (Figure 2A). As a positive and negative control, ACE bound to glutathione S-transferase (GST)-PvERF15 but not the GST mock [9]. We observed that ACE is able to be bound by nuclear proteins (Figure 2B). The ACE–protein complex was significantly inhibited by the addition of a 50-fold molar excess of unlabeled 2 × ACE (Figure 3A, lanes 1 and 8). We therefore conclude that ACE specifically binds to the nuclear proteins in vitro.

### 2.3. The First 9-bp Fragment of ACE Is Required by Nuclear Protein Binding

To further identify the specific sequence of ACE for nuclear protein binding, a competitive EMSA assay was employed. A series of three-base sequential substitution ACE versions (ACE-M1 to -M6) was used as the competitors (Figure 3A). ACE-M2, -M4, -M5, and -M6 were able to completely inhibit the interaction of nuclear proteins with the ACE probe. However, ACE-M1 and ACE-M3 showed a relatively weaker inhibitory effect on nuclear protein binding. Thus, the 1–3 and 6–9 base pairs of ACE appear essential for nuclear protein binding. We named this first 9-bp sequence (5′- CCTRRRCCC-3′, R = A/G) as ACE_core_.

Since ACE_core_ produced a similar shifted band to that produced by ACE (Figure 3B, lanes 1 and 3), ACE_core_ and ACE seem to interact with the same and/or similar DNA-binding proteins. An excessive amount of ACE_core_ (Figure 3B, lanes 4 and 5), but not mutated ACE_core_ (mACE_core_, Figure 3B, lanes 6 and 7), inhibited the binding of ACE_core_. Taken together, we conclude that ACE_core_ is sufficient to specifically bind with nuclear proteins.

### 2.4. PvMTF-1 and PvERF15 Respond to Ethylene Stress in Bean Seedlings

Since ACE serves as an ERE in *Arabidopsis*, we investigated whether ACE-containing *PvMTF-1* and its transcription regulator PvERF15 are regulated by ethylene in the etiolated bean seedlings (Figure 4A). Treatment with ACC significantly increased the mRNA levels of both *PvMTF-1* and *PvERF15* (Figure 4B), suggesting that *PvMTF-1* and *PvERF15* responded to ethylene stress. Given that PvERF15 is located the upstream of *PvMTF-1* and acts as a transcription regulator of *PvMTF-1* via its ACE [8], ACE is involved in ethylene-induced *PvMTF-1* expression.

## 3. Discussion

### 3.1. Our Findings Provide Evidence for ACE as a Novel ERE in Arabidopsis

In this study, we provided evidence supporting ACE as an ERE in *Arabidopsis* seedlings. In the transient *GUS* assay, ACE was able to confer ethylene stress responsiveness on a minimal promoter, whereas the mutated ACE was not (Figure 1C). When the endogenous ethylene perception was blocked by Ag^+^, ACE-mediated GUS expression was also inhibited (Figure 1C). This reflects the effects of ethylene-induced inhibition of ACE-mediated *GUS* expression. These findings suggested that ACE-mediated GUS expression is regulated by ethylene stress and ethylene action and, thus, ACE acts as an ERE in *Arabidopsis*. This can further be supported by the observation that ACE serves as a *cis*-acting element to specifically bind with nuclear proteins (Figure 2B). In this study, genetic evidence for ACE as an ERE is mainly based on *GUS* transient assay. To obtain a more complete understanding of ACE function in ethylene response, stably transformed *Arabidopsis* lines will be needed for future studies.

Given that the ACE_core_ sufficiently supports the ACE binding with the nuclear proteins (Figure 3A,B), ACE_core_, to some extent, may represent ACE. To assess the ACE_core_ distribution in the promoter regions (−1000 bp) of *Arabidopsis* genes, a genome-wide survey was performed. We found that 847 *Arabidopsis* genes contain 1007 ACE_core_ in their promoters (Appendix A). ACE_core_ appears to be able to globally regulate gene expression in *Arabidopsis*. However, gene ontology (GO) term enrichment analysis found no significant enrichment for 847 ACE_core_-containing genes. In the future, further assessment of the regulatory role of ACE_core_ in these genes could elucidate the biological significance of ACE.

### 3.2. The Biological Implication of ACE in the Ethylene Induction of PvMTF-1

Our previous study demonstrated that ACE acts as a positive element through which PvERF15 activates *PvMTF-1* expression [8]. In this study, we showed that ethylene stress transcriptionally induced the expression of *PvERF15* and *PvMTF-1* in bean seedlings (Figure 4B). Additionally, other known ERF-binding sites [12] were not found in the 397-bp *PvMTF-1* promoter [8,13], implying that ACE seems to be a unique ERE for *PvMTF-1*. Given this, we propose that upon ethylene stress, *PvERF15* expression is induced and then enhances *PvMTF-1* expression via directly binding on ACE. Although we cannot exclude the other transcription regulators of *PvMTF-1* ethylene-induced expression, ACE interaction with PvERF15 may contribute to the ethylene induction of *PvMTF-1,* and thus demonstrate the biological implication of ACE in the ethylene induction of *PvMTF-1*.

In conclusion, the GCC box and ACE share the features of ERF binding and ethylene response. However, in contrast to the high ethylene-inducible ability of the GCC box [4], ACE has a low induction ability (about two-fold induction). Despite this, our findings provide evidence for a new ERE and extend our knowledge of ethylene-regulated transcription.

## 4. Materials and Methods

### 4.1. Plant Materials and Growing Conditions

Seeds of *Arabidopsis thaliana* (Columbia-0) or bean (*Phaseolus vulgaris*) were surface-sterilized. *Arabidopsis* seeds were grown on 1/2 MS agar plates without (as a control) or with 10 μM ACC or 50 μM AgNO_3_ in the dark (wrapped in aluminum foil) at 22 °C for 14 d. The etiolated *Arabidopsis* seedlings were subjected to hypocotyl length analysis, root length analysis, transient expression assay, or nuclear protein extraction. Bean seeds were grown on MS agar plates without (as a control) or with 500 μM ACC in the dark at 22 °C for 10 d. The etiolated bean seedlings were subjected to RNA isolation.

### 4.2. Agrobacterium Tumefaciens-Mediated Gene Transfer by Infiltration of Arabidopsis Seedlings

*ACE-35S_(-46/+8)_:GUS*, *ACEm-35S_(−46/+8)_:GUS*, or *35S:GFP* (pCAMBIA1302) was used for the transient expression assay as described previously [8]. The etiolated *Arabidopsis* seedlings were vacuum-infiltrated with *A. tumefaciens* as described previously by Marion et al. [14]. Briefly, *A. tumefaciens* cells were collected and resuspended at an optical density (OD 600 nm = 2.0) in 1/4 MS liquid medium containing 5% (*w*/*v*) sucrose, 200 mM acetosyringone, and 0.01% (*v*/*v*) Silwet. After vacuum infiltration, transformed *Arabidopsis* seedlings were transferred to filter paper soaked with 1/4 MS solution medium with or without 10 μM ACC or 50 μM AgNO_3_ in the dark (wrapped with aluminum foil) at 22 °C for 2 d.

### 4.3. Protein Extraction

Total protein from the plant samples was extracted using extraction buffer (50 mM Tris-HCl) pH 7.5), 150 mM NaCl, 1 mM EDTA, 10% (*v*/*v*) glycerol, 5 mM dithiothreitol, 0.5% (*v*/*v*) Triton X-100, 1 mM phenylmethylsulphonyl fluoride, and 1% (*v*/*v*) Nonidet P-40).

Nuclear fractionation was performed based on the protocol described by Xia et al. [15] with some modifications. Briefly, 2 g of etiolated *Arabidopsis* seedlings was homogenized in Honda buffer (2.5% Ficoll 400, 5% dextran T40, 0.4 M sucrose, 25 mM Tris-HCl, pH 7.4, 10 mM MgCl_2_, 10 mM β-mercaptoethanol, and a proteinase inhibitor cocktail) using a mortar and pestle. The homogenate was filtered through two layers of 64-μm (pore-size) nylon mesh. Triton X-100 was added to a final concentration of 0.5%, and the mixture was incubated on ice for 15 min. The mixture was centrifuged at 1500× *g* for 5 min. The supernatants were used as the cytosolic protein, and the pellets were washed twice in Honda buffer containing 0.1% Triton X-100 and used as the nuclei-enriched fraction. This nucleus-enriched preparation was centrifuged at 100× *g* for 5 min to pellet the starch and cell debris. The supernatant was centrifuged subsequently at 2000× *g* for 5 min to pellet the nuclei. The purified nuclei were resuspended gently in 125 μL of Honda buffer and transferred to a microcentrifuge tube.

GST-PvERF15 fusion protein or GST protein (mock) was prepared as previously described [8].

### 4.4. EMSA

The 5′ biotin-labeled DNA probes or the corresponding unlabeled DNA were synthesized by Sangon Biotechnology (Shanghai, China). The top strand and the bottom strand were responded in the annealing buffer (50 mM Tris-HCl (pH 7.5), 250 mM NaCl, 0.5 mM EDTA) at a 1:1 ratio. Heat the mixture to 95 °C for 4 min to remove all secondary structures, and then anneal it at 60–70 °C for 20 min. Following this, gradually decrease (such as 5 °C/min) the temperature to 25 °C and maintain this for 1–2 h. EMSA was performed using the LightShift Chemiluminescent EMSA Kit (Pierce) according to the manufacturer’s instructions. A total of 100 ng of nuclear proteins was incubated together with 1.5 ng of biotin-labeled probes in 25-μL reaction mixtures (1× binding buffer, 50 ng poly(deoxyinosinic-deoxycytidylic acid), 2.5% (*v*/*v*) glycerol, 0.05% (*v*/*v*) Nonidet P-40, and 5 mM MgCl_2_) for 30 min at room temperature. For the cold competitor, the unlabeled competitors were added into the reaction mixture. The reaction mixtures were separated on 6% (*w*/*v*) native PAGE gels. The labeled probes were detected according to the instructions provided with the EMSA kit using a Fujifilm LAS-3000 imager.

### 4.5. Western Blot Analysis

Proteins separated on a gel were electrophoretically transferred to a pure nitrocellulose blotting membrane (Pall Life Sciences). The membrane was cut across the molecular mass region of the corresponding proteins and separately probed with an anti-GUS antibody (ab50148, Abcam), an anti-GFP antibody (ab290, Abcam), or anti-tubulin antibody (T5168, Sigma-Aldrich), and an anti-histone H3 antibody (AS10 710, Agrisera) or an anti-UGPase antibody (AS05 086, Agrisera). Protein blots were developed with an ECL kit (Amersham Pharmacia Biotech), and images were obtained using the LAS3000 image-capture system (Fujifilm). The Adobe Photoshop CS histogram analysis tool was used for the grayscale analysis of the Western blot bands of GUS and GFP.

### 4.6. RNA Isolation and cDNA Synthesis

Total RNA was extracted from the etiolated bean seedlings using the RNAprep pure plant kit with on-column DNase digestion (Tiangen Biotech, China). RNA (about 2 μg) was used to synthesize the first-strand cDNA with oligo(dT) primers with the PrimeScript first-strand cDNA synthesis kit (Takara).

### 4.7. Real-Time Quantitative Reverse Transcription (qRT)-PCR Analysis

The qRT-PCR analysis of *PvERF15* (accession number: XM_007144842) or *PvMTF-1* (accession numbers: DQ109993 and U54704) was performed on a the CFX96 Real-Time System (Bio-Rad) with the SYBR Premix Ex-Taq Kit (Takara) according to the manufacturer’s instructions. The bean *ACTIN* gene (accession number: AB067722) was used as an internal control. The thermal cycling conditions were 95 °C for 3 min followed by 50 cycles of 95 °C for 30 s, 55 °C for 30 s, and 72 °C for 30 s. The primers used for qRT-PCR were as follows: PvERF15-F (5′-TGGCACTCAAGAGGAAACACAC-3′) and PvERF15-R (5′-CATTCTCCAACTGGTGCTCCC-3′) for *PvERF15*, PvMTF-1-F (5′-CTGAAGAGTTGCGAATTGCACGAA-3′) and PvMTF-1-R (5′-TTGAATTTGATTGAATCTTGCAGGATG-3′) for *PvMTF-1*, and Actin-F (5′-CACCGAGGCACCGCTTAATC-3′) and Actin-R (5′-CGGCCACTAGCGTAAAGGGAA-3′) for *ACTIN*. The relative expression levels were analyzed using a delta–delta cycle threshold method.

### 4.8. Bioinformatics Analysis

Using CCTRRRCCC (R = A or G) as a query, the Patmatch program available at the TAIR web site (http://www.arabidopsis.org/cgi-bin/patmatch/nph-patmatch.pl) was employed to search the ACE_core_-containing genes based on TAIR10 Loci Upstream Sequences (−1000 bp). The GO Term Enrichment for Plants tool (https://www.arabidopsis.org/tools/go_term_enrichment.jsp) was used for 847 ACE_core_-containing genes.

### 4.9. Statistical Analysis

Statistical analysis (SE and *p*-values) was conducted using Microsoft Excel 2007. Significance (*p* < 0.05) was assessed using Student’s *t*-tests.

## Figures and Tables

**Figure 1 plants-09-01033-f001:**
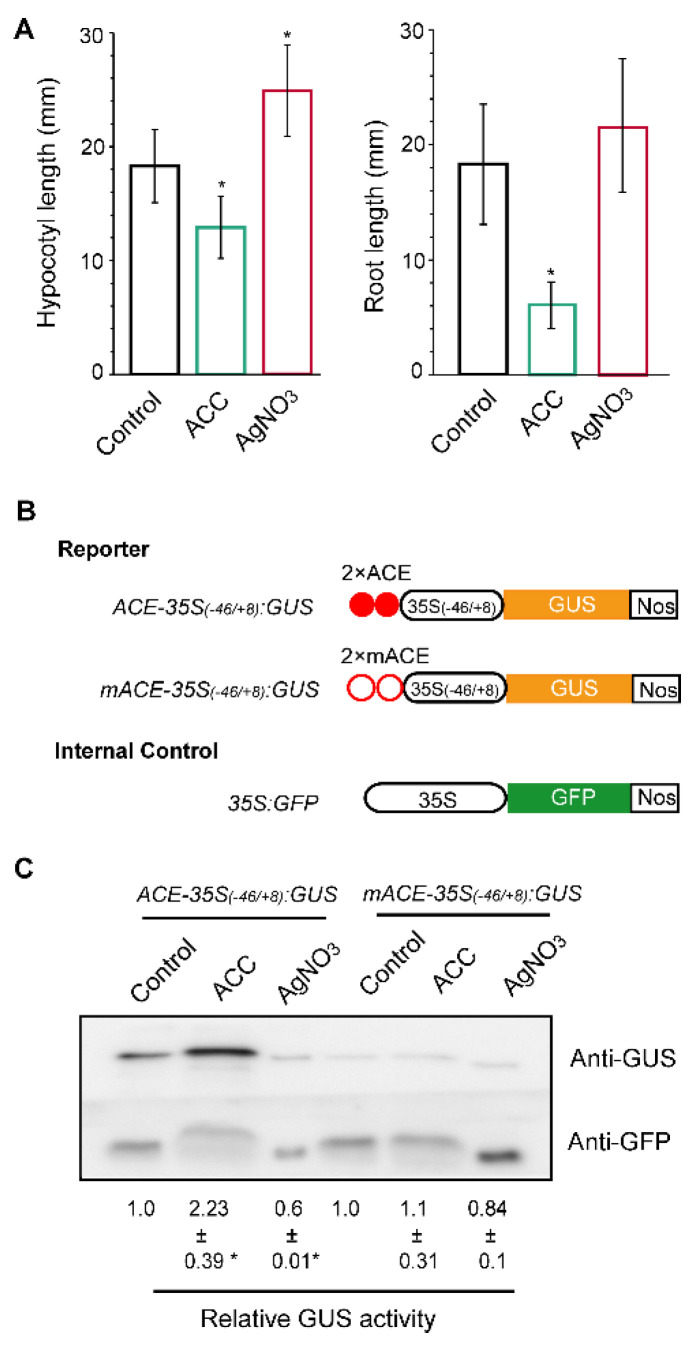
GUS transient assays for ACE as an ERE in the etiolated *Arabidopsis* seedlings. (**A**) The *Arabidopsis* seeds were grown in 1/2 glutathione S-transferase (MS) without (control) and with 10 μM ACC (ACC) or 50 μM AgNO_3_ (AgNO_3_) in the dark for two weeks and then subjected to hypocotyl or root length analysis. Data are means ± standard error (SE) (*n* = 100). The significance was assessed using a one-sided Student’s *t*-test (* *p* ≤ 0.05). (**B**,**C**) *GUS* transient assays in the etiolated *Arabidopsis* seedlings. Schematic diagram of constructs used in the experiments (**B**). The etiolated *Arabidopsis* seedlings were co-transformed with reporter and internal control constructs. The transformed etiolated *Arabidopsis* seedlings were grown on the solution medium without (control) or with 10 μM ACC (ACC) or 50 μM AgNO_3_ (AgNO_3_) for 2 days. The expression of *GUS* or *GFP* was determined by immunoblot using an anti-GUS antibody and anti-GFP antibody from the same protein samples. (**C**) Relative *GUS* activity was measured as the relative band intensity of GUS to GFP (the control was set as 1). The data represent an average from two independent experiments. Means ± SE are shown. The significance was assessed by a one-sided Students *t*-test (* *p* ≤ 0.05).

**Figure 2 plants-09-01033-f002:**
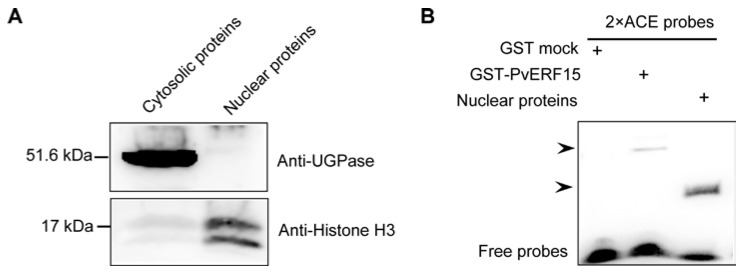
ACE specifically binds to the nuclear proteins in vitro. (**A**) Western blot verified the successful isolation of nuclear proteins using an anti-histone H3 (nuclei marker) antibody and anti-glucose pyrophosphorylase (UGPase) (cytoplasm marker) antibody. (**B**) EMSA was performed using biotin-labeled 2 × ACE probes with the affinity-purified GST-PvERF15 (positive control), GST mock (negative control), and nuclear proteins from the etiolated *Arabidopsis* seedlings. The bound complex is indicated by the arrows. The experiments were performed two times showing similar results.

**Figure 3 plants-09-01033-f003:**
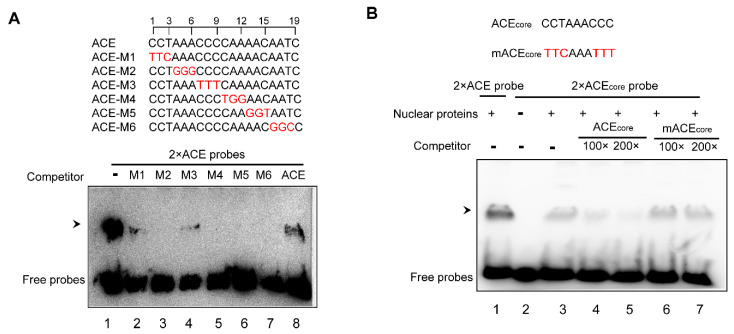
Competitive EMSA identifies the required sequences of ACE for nuclear protein binding. (**A**) ACE (2×) was used as the probe in the EMSA with the nuclear proteins, and the mutant variants (red cases) were used as competitors (50-fold molar). (**B**) Gel shifts with the ACE_core_ probe (2×) showing specific binding to the same and/or similar nuclear proteins as ACE. Mutations introduced into the ACE_core_ are shown in red. A repeat experiment confirmed the results (Appendix A).

**Figure 4 plants-09-01033-f004:**
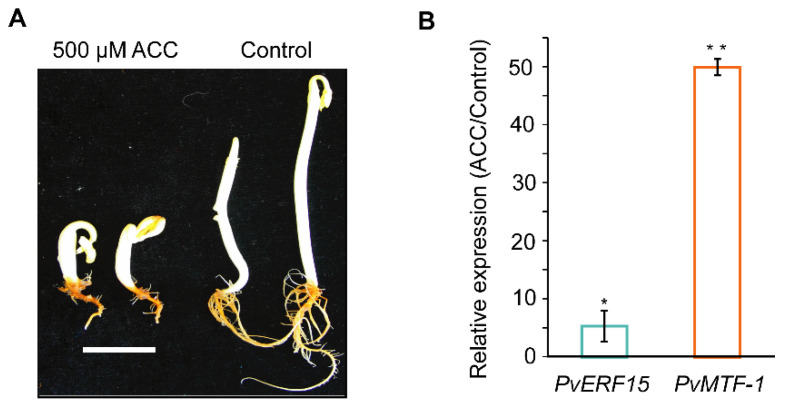
ACC regulates the mRNA levels of both *PvMTF-1* and *PvERF15* in bean seedlings. (**A**) Bean seeds were germinated and grown on the MS without (control) or with 500 μM ACC for 10 days in the dark. ACC induced a typical ethylene-responsive phenotype, indicating the effects of ethylene stress. Bar = 1 cm. (**B**) The etiolated bean seedlings were subjected to quantitative reverse transcription qRT-PCR analysis. ACC-induced expression level of *PvERF15* or *PvMTF-1* was expressed as a ratio relative to control, which was set to a value of 1. Data shown are averages of three independent qRT-PCR experiments. Error bars represent SEs. Significance between experimental values was assessed using one-sided Student’s *t* test (*, *p* ≤ 0.05, and **, *p* ≤ 0.01).

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
