# Peer review of "An AC-Rich Bean Element Serves as an Ethylene-Responsive Element in Arabidopsis"

_plants, 2020, doi:10.3390/plants9081033_

Round 1
Reviewer 1 Report
In this paper, the authors are functionally analyzing a new cis element (ACE) previously found in the promoter of Pv MTF-1 gene. The general claim in the paper is that this ACE behaves as an ethylene-responsive element.
The paper counts 4 main figures and one supplementary table. I was unable to have access to sup table 1 and this has thus not been reviewed. The paper is short, very concise and the figures of good quality. The data presented fully support the general claim.
My main concern is about the material and method, which is also very concise and would need some additional details, even if authors have performed the experiences “according to the instruction manual”.
For example: the relative GUS activity given on figure 1 (line 61 and following) is not explained in the Mat & Meth section. I presume this is based on image quantification but please indicate software used, type of device etc… Usually, GUS expression quantification is made using fluorometry measurements, which is more precise than Western blot quantification. Why have the authors not chosen fluorometry-based quantification?
1- On figure 1-B please indicate what mutation was made (mACE 35S -46/+8), in reference to mutations presented on figure 3, if relevant.
2- On figure 1C, the 2 controls are differentially accumulated, which suggests that the minimal promoter with the ACE motif is constitutively expressed with some background. Please comment.
3- Figure 3: please add a second set of experiments with similar results as sup data.
4- Line 149: I do not understand the sentence “therefore ACE core may represent ACE”. Please rephrase.
5- Line 149-152: I had no access to sup figure 1. Nevertheless, have the authors checked if the 847 Ath genes containing ACEcore cis element are grouped according to their Gene Ontology or linked to specific metabolic pathways? This would indicate possible co regulation of the genes containing this cis element?
6- Line 202: EMSA: please give indications about the buffer used (as often critical), the quantities used on the gels …. As it is clearly explained for protein extraction. Here we cannot understand how the experience was really done and tis figure is central in the paper.
7- Line 206: western blot: please give the reference number of the antibody used, not only the provider. figure 3: please
Reviewer 2 Report
In my opinion, this is an excellent research that deserves to be published in its present form. I consider the research displays the originality enough to be fastly published in this Journal.
Author Response
In my opinion, this is an excellent research that deserves to be published in its present form. I consider the research displays the originality enough to be fastly published in this Journal.
Author Response: We appreciate this very positive comment on our work. We thank the reviewer for taking the time to review our manuscript.
Reviewer 3 Report
I thoroughly read the manuscript entitled ‘A bean AC-rich element serves as an ethylene-responsive element in Arabidopsis’. The work is well written, executed and thought. I am sure this will provide scientific community to look for this motif in known genes downstream to ethylene and It will serve a part of repository of novel cis elements. Manuscript can be accepted in its present form.
Author Response
I thoroughly read the manuscript entitled ‘A bean AC-rich element serves as an ethylene-responsive element in Arabidopsis’. The work is well written, executed and thought. I am sure this will provide scientific community to look for this motif in known genes downstream to ethylene and It will serve a part of repository of novel cis elements. Manuscript can be accepted in its present form.
Author Response: We thank the reviewer for this very positive evaluation of our work. We thank the reviewer for taking the time to review our manuscript.